Biogas slurry application alters soil properties, reshapes the soil microbial community, and alleviates root rot of Panax notoginseng

Wang Chengxian 1 2
Liu Jianfeng 1 2 3
Wang Changmei 1 2 3
Zhao Xingling 1 2 3
Wu Kai 1 2 3
Yang Bin 1 2 4
Yin Fang 1 2 3
Zhang Wudi 2071@ynnu.edu.cn wootichang@163.com 1 2 3
1 Engineering and Research Center of Sustainable Development and Utilization of Bioenergy, Ministry of Education, Yunnan Normal University , Kunming , China
2 Yunnan Research Center of Biogas Technology and Engineering, School of Energy and Environment Science, Yunnan Normal University , Kunming , China
3 Jilin Dongsheng Institute of Biomass Energy Engineering , Tonghua , China
4 Graduate School, Yunnan Normal University , Kunming , China
LaMontagne Michael
Electronic publication date: 2022 Jul 26
Publication date: 2022
Volume: 10
Electronic Location ID: e13770
Received 2021 Dec 14; Accepted 2022 Jul 1
Copyright: ©2022 Wang et al.
Copyright year: 2022
Copyright holder: Wang et al.
License: This is an open access article distributed under the terms of the Creative Commons Attribution License, which permits unrestricted use, distribution, reproduction and adaptation in any medium and for any purpose provided that it is properly attributed. For attribution, the original author(s), title, publication source (PeerJ) and either DOI or URL of the article must be cited.
License URL: https://creativecommons.org/licenses/by/4.0/

Keywords: Biogas slurry, Panax notoginseng, Root rot disease, Soil physicochemical property, Microbiota

Funding: Yunnan Biogas Engineering and Resource Utilization Model Worker Innovation Studio Yunnan Ten Thousand Talents Plan Industrial Technology Champion Project 20191096 Science and Technology Development Project of Jilin Province 20200402099NC 20200403010SF This work was supported by the Yunnan Biogas Engineering and Resource Utilization Model Worker Innovation Studio, Yunnan Ten Thousand Talents Plan Industrial Technology Champion Project (20191096), and the Science and Technology Development Project of Jilin Province (20200402099NC and 20200403010SF). The funders had no role in study design, data collection and analysis, decision to publish, or preparation of the manuscript.

==============================
Background

Panax notoginseng is an important herbal medicine in China, where this crop is cultivated by replanting of seedlings. Root rot disease threatens the sustainability of P. notoginseng cultivation. Water flooding (WF) is widely used to control numerous soilborne diseases, and biogas slurry shows positive effects on the soil physiochemical properties and microbial community structure and has the potential to suppress soilborne pathogens. Hence, biogas slurry flooding (BSF) may be an effective approach for alleviating root rot disease of P. notoginseng; however, the underlying mechanism needs to be elucidated.

Methods

In this study, we conducted a microcosm experiment to determine if BSF can reduce the abundance of pathogens in soil and, alleviate root rot of P. notoginseng. Microcosms, containing soil collected from a patch of P. notoginseng showing symptoms of root rot disease, were subjected to WF or BSF at two concentrations for two durations (15 and 30 days), after which the changes in their physicochemical properties were investigated. Culturable microorganisms and the root rot ratio were also estimated. We next compared changes in the microbial community structure of soils under BSF with changes in WF and untreated soils through high-throughput sequencing of bacterial 16S rRNA (16S) and fungal internal transcribed spacer (ITS) genes amplicon.

Results

WF treatment did not obviously change the soil microbiota. In contrast, BSF treatment significantly altered the physicochemical properties and reshaped the bacterial and fungal communities, reduced the relative abundance of potential fungal pathogens (Fusarium, Cylindrocarpon, Alternaria, and Phoma), and suppressed culturable fungi and Fusarium. The changes in the microbial community structure corresponded to decreased root rot ratios. The mechanisms of fungal pathogen suppression by BSF involved several factors, including inducing anaerobic/conductive conditions, altering the soil physicochemical properties, enriching the anaerobic and culturable bacteria, and increasing the phylogenetic relatedness of the bacterial community.

Conclusions

BSF application can reshape the soil microbial community, reduce the abundance of potential pathogens, and alleviate root rot in P. notoginseng. Thus, it is a promising practice for controlling root rot disease in P. notoginseng.

Introduction

Biogas production, through anaerobic digestion of human and animal waste, straw, and other organic materials, has emerged as a potentially important source of renewable energy (Kougias & Angelidaki, 2018). In recent decades, the number of biogas plants in China and other countries has greatly increased (Kougias & Angelidaki, 2018). The rapid development of biogas production has led to increases in the amount of biogas residue, including that of biogas slurry (BS). BS is a high-quality organic fertilizer with high levels of residual organic carbon, available nitrogen, and other minerals useful for crop growth (Abubaker, Risberg & Pell, 2012; Insam, Gómez-Brandón & Ascher, 2015). Hence, application of BS alters the soil microbial community structure, increases soil microbial diversity and activity, improves soil quality, and enhances crop yield (Abubaker, Risberg & Pell, 2012; Cristina et al., 2020; Walsh et al., 2012). Furthermore, BS contains numerous organic compounds (such as volatile fatty acids (VFAs)) and a high concentration of ammonia, which can suppress Fusarium spp. and other soil pathogens (Cao et al., 2014; Huang et al., 2015b). Consistent with the effects of other organic amendments, BS application can suppress soilborne pathogens and reduce the incidence of plant diseases (Cao et al., 2016; Insam, Gómez-Brandón & Ascher, 2015). For example, Cao et al. (2016) showed that application of BS, under both normal moisture and flooding conditions (BSF), suppressed Fusarium wilt disease in watermelon. Appropriate application of BS to soil contributes to the decreased use of pesticides and chemical fertilizers, demonstrating the potential of BS in sustainable agriculture production and environmental protection (Insam, Gómez-Brandón & Ascher, 2015; Walsh et al., 2012).

Panax notoginseng (Burk.) F. H. Chen (“Sanqi” in Chinese) is a member of the family Araliaceae and is one of the most important herbal medicines in China. Wenshan County in Yunnan province (southwest China) is the geo-authentic habitat for Sanqi planting. Sanqi requires a specific ecological habitat and is grown using a continuous monoculture system. This leads to a high prevalence of replanting failure. Therefore, the planting area has been extended to the surrounding areas and beyond, which has further negatively affected the yield and quality of Sanqi and led to the development of more serious diseases, such as mildew and root rot. Abiotic and biotic factors, including the deterioration of soil physicochemical characteristics, nutrient imbalance, soilborne diseases, and accumulation of phytotoxic allelochemical substrates cause replanting failure (Liu, Yang & Zhu, 2019; Wu et al., 2008). Root rot is the most common fungal disease associated with soilborne pathogens, such as Fusarium, Alternaria, Cylindrocarpon, and Phoma (Li et al., 2020; Miao et al., 2006; Wang et al., 2021). Among these, Fusarium spp. are the primary pathogens because of their wide host range and high-stress tolerance (e.g., drought and high temperatures), making them difficult to control (Liu, Yang & Zhu, 2019). Different measures have been explored for mitigating diseases in Sanqi, among which, rotation (with maize, rape, and wheat) is a preferred practice (Liu, Yang & Zhu, 2019; Tang et al., 2020). However, even after 10–20 years, rotation cannot eliminate pathogens (Tang et al., 2020). Additionally, fungicides are not environmentally benign and are gradually becoming unavailable because of the strict regulation of agrochemicals. Hence, more effective and non-chemical biological measures, such as soil flooding and/or addition of organic materials (straw, BS), are urgently needed. Water flooding (WF), an ancient and widely used practice in China and other Asian countries, effectively controls numerous soilborne diseases (Niem, Gundersen & Inglis, 2013). BSF, which incorporates organic materials and associated microorganisms, can increase soil microbial activity, and promote nutrient availability (Cao et al., 2016; Dahunsi et al., 2021). BSF also has positive effects on the soil physiochemical properties (such as increasing the soil pH, available K contents, NH4 +-N contents, water-soluble carbon, and water-soluble nitrogen) and microbial community structure (suppression of soilborne pathogens), BSF can alleviate plant diseases (Cao et al., 2014; Cao et al., 2016), introduces organic materials into soil (Abubaker, Risberg & Pell, 2012; Insam, Gómez-Brandón & Ascher, 2015), and rapidly creates reductive/anaerobic conditions (Cao et al., 2016). In this manner BSF is similar to anaerobic soil disinfestation (ASD). ASD involves flooding soil after the addition of organic residues (Blok et al., 2000) and has been widely used for effective disinfestation of various soilborne pathogens (Hewavitharana & Mazzola, 2016; Strauss & Kluepfel, 2015; Zhou et al., 2019).

Studies of the changes in the soil physiochemical properties and microbial community under BSF and/or WF treatments for Sanqi root rot disease, particularly studies aimed at evaluating changes in pathogen abundance are lacking. We predicted that compared with WF, BSF can more effectively suppress pathogens and alleviate root rot disease in P. notoginseng. Here, we conducted a microcosm experiment using Sanqi root rot soil, which was treated with WF and two concentrations of BSF. We (1) explored the changes in the physicochemical properties and composition of microbial community in the treated soils, (2) assessed the efficacy of WF and BSF in suppressing pathogens and alleviating root rot symptoms, and (3) examined the mechanisms underlying these responses.

Material and Methods

Biogas slurry and soil characteristics

BS was collected from an internal-circulation biogas reactor, using vegetable juice waste as input material, at Bio-energy and Environment Engineering Research Group, Yunnan Normal University, Kunming, China. The reactor had been stably operated for 6 months. The chemical characteristics of the BS were as follows: total solid 1.0 ± 0.2%, chemical oxygen demand 7,072 ± 65 mg/L, total N 612.2 ± 22.4 mg/L, NH4+-N 282.7 ± 14.6 mg/L, NO3−-N 49.6 ± 5.3 mg/L, and pH 6.5 ± 0.2. Bulk soil samples were collected from a Sanqi plantation in Wenshan (23° 40′N, 102° 35′E, 1,400 m altitude), Yunnan Province, China, in January 2019. This site is classified as Latosols based on the Chinese soil taxonomy. The region is characterized by a subtropical climate, with a mean annual precipitation and temperature of 1,100–1,319 mm and 15–18 °C, respectively. Sanqi has been consecutively cultivated for 5 years and suffered severe root rot disease at this site. Fusarium spp. have been frequently isolated and identified as the main pathogens underlying this disease. The chemical characteristics of the initial soil were as follows: pH 6.8 ± 0.2, organic matter 19.5 ± 1.7 g/kg, total N 925.5 ± 43.2 mg/kg, available P 66.0 ± 4.5 mg/kg, and available K 94.4 ± 7.1 mg/kg. Approximately 10 kg of continuous-cropping soil was randomly collected from the 0–20 cm soil layer, homogeneously mixed, and sieved through a 2-mm mesh to remove stones and plant debris.

Experimental design and soil sampling

A soil microcosm experiment was conducted to investigate the effects of WF and BSF treatments on the soil physicochemical properties and microbiota. Three treatments (1:1 soil: water ratio was selected according to a previous study (Wen et al., 2015)), each with three replicates, were performed in 500-mL microcosms (glass bottles) as follows: (1) water flooding treatment (CK): 250 g soil flooded with 250 mL sterilized water; (2) diluted-BS treatment (CH): 250 g soil flooded with 250 mL diluted 50%-concentration BS (equivalent to ∼0.3 g N/kg soil); and (3) original BS treatment (CF): 250 g soil flooded with 250 mL original BS (equivalent to ∼0.6 g N/kg soil). Specifically, after adding BS at two concentrations to the initial soil (untreated soil, named as “Soil”), the homogeneously mixed slurry samples were immediately collected (named as “CH0d” and “CF0d” respectively, representing two BS-flooded soils without incubation), and then the glass bottles were sealed and incubated in the dark at 28 °C. WF treatment was performed using a similar procedure. After incubation for 15 and 30 days, day 15 (CK15d, CH15d, and CF15d, representing water and two BS-flooded soils with 15 day’s incubation) and day 30 soil samples (CK30d, CH30d, and CF30d, representing water and two BS-flooded soils with 30 day’s incubation) were collected (∼20 g per sample). Thus, a total of 27 samples was obtained. Each sample was divided into two parts: one part was used to analyze the soil physicochemical properties and assess the number of culturable bacteria and fungi, and the other was stored at −80 °C for subsequent DNA extraction.

Analysis of soil physicochemical properties

Soil pH and oxidation–reduction potential (Eh) were determined using a PHS-3C Meter with the corresponding electrodes (INESA Scientific Instrument Co., Ltd., Shanghai, China) and a 1:2.5 soil/water (w/v) suspension. Electrical conductivity (EC) was measured in a 1:5 soil/water (w/v) suspension using a DDS-11A Conductivity Meter (INESA Scientific Instrument Co., Ltd., Shanghai, China). Ammonia nitrogen was measured with a continuous flow analyzer (FIAstar TM 5000 System; FOSS, Hilleroed, Denmark). Potential toxic organic acids (mainly VFAs, including acetate, propionate, butyrate, and valerate) were analyzed using gas chromatography on a GC-9790II (FULI Apparatus Co. Ltd., Shanghai, China), as previously described (Zhao et al., 2018). All values were obtained from three replicates in each treatment.

Assay of culturable bacteria, fungi, and Fusarium

To assess the effects of water or BS treatments on the microbial community in the soils, we determined the population densities of culturable bacteria, fungi, and Fusarium (a potential pathogen) using a standard 10-fold dilution plating assay. Briefly, the bacteria on nutrient agar (NA) medium plates were counted, and colony-forming units (CFUs) were counted after incubation at 30 °C for 2 days. Fungi and Fusarium were enumerated using Martin’s Rose Bengal agar and Komada’s selective medium (Komada, 1975), respectively, after incubation at 25 °C for 5 days. All values were obtained from three replicates.

Pathogenicity assay of soil on Sanqi root

A pathogenicity assay (potential to cause root rot) of the water or BS-treated soils described above was performed on Sanqi roots in vitro according to a previous method (Luo et al., 2019) with some modifications. Briefly, on day 30 of the experiment, surface water was removed and the remaining treated soils were air-dried for 4 days (approximately 40% water holding capacity) and then thoroughly mixed. Healthy 1-year-old roots were washed with sterile water, then surface sterilized with 1% sodium hypochlorite for 6 min, and washed four times with sterile water. The roots were transferred to a plastic container containing the treated soils and were completely covered with the soils. Each treatment (CK, CH, and CF) and the untreated soil contained three replicates (each with 10 roots). All treatments were randomly placed in an incubator at 25 °C and watered every 3 days to maintain the water holding capacity at approximately 40%. After 30 days, the root rot ratio (%) was calculated as the number of roots showing rot-rooted symptoms divided by the total number of tested roots.

Soil DNA extraction and sequencing

Total genomic DNA was extracted from each sample using a PowerSoil® DNA Isolation Kit (MoBio Laboratories, Carlsbad, CA, USA). The DNA concentration and quantity were evaluated using a NanoDrop ND-1000 spectrophotometer (Thermo Fisher Scientific, Waltham, MA, USA); the extracted DNA was stored at −20 °C until use. Marker genes, amplified using polymerase chain reaction (PCR), were sequenced to characterize the community composition and diversity of bacteria and fungi. The prokaryotic 16S V3-V4 and fungal internal transcribed spacer (ITS2) regions were amplified using the primer pairs 341F (CCTAYGGGRBGCASCAG)/806R (GGACTACNNGGGTATCTAAT) and ITS3-2024F(GCATCGATGAAGAACGCAGC)/ ITS4-2409R (TCCTCCGCTTATTGATATGC), respectively (Takahashi et al., 2014; Toju et al., 2012). PCR was performed using Phusion® High-Fidelity PCR Master Mix (New England Biolabs, Ipswich, MA, USA). The amplification conditions were as follows: initial denaturation at 95 °C for 2 min, followed by 25 cycles of denaturation at 95 °C for 30 s, annealing at 55 °C for 30 s, and extension at 72 ° C for 1 min, with a final extension at 72 °C for 10 min. The PCR products were purified with a GeneJET™ Gel Extraction Kit (Thermo Fisher Scientific) and used to construct sequencing libraries using an Ion Plus Fragment Library Kit (Thermo Fisher Scientific), according to the manufacturer’s recommendations. Library quality was assessed using a Qubit@ 2.0 Fluorometer (Thermo Fisher Scientific). The library was sequenced on an Ion S5™ XL platform and 600 bp single-end reads were generated. The sequencing data generated was deposited in the NCBI Sequence Read Archive database (accession numbers PRJNA661430 and PRJNA661668 for the bacterial and fungal sequences, respectively).

Bioinformatics analyses

Raw sequencing reads were filtered and analyzed using the QIIME software (v1.9.1) (Caporaso et al., 2010). Briefly, primer sequences and low quality reads with scores below Q30 were filtered, and chimeras were detected using Chimera UCHIME. Filtered high-quality sequences showing ≥ 97% similarity were clustered and assigned to the same operational taxonomic unit (OTU) using Uparse software (v7.0.1001) (Edgar, 2013). To obtain taxonomic information on the bacterial and fungal OTUs, a representative sequence of each OTU was generated and aligned against the Silva (v132) and Unite (v7.2) databases, respectively. The OTU abundance was normalized using a standard sequence number corresponding to each sample with the fewest sequences (45,000 sequences for bacteria and 32,000 for fungi).

Alpha diversity indices, including the observed-species and Shannon indices, were determined to analyze within-sample diversity. Beta diversity analyses were performed to evaluate differences between samples (treatments). Principal coordinate analysis (PCoA) and hierarchical cluster analysis were both based on Bray–Curtis distance using an OTU abundance table, and then visualized using “ggplot2” (v3.3.3) and “ggtree” (v2.2.1) (Yu et al., 2017) in R (v3.5.1), respectively. Linear discriminant analysis (LDA) coupled with effect size was performed using LEfSe software (Segata et al., 2011), and the default LDA score was 3.5. Metastats analysis was performed using permutation tests between groups at the genus level to obtain the P values (adjusted with the “false discovery rate” method) (Ouzounis et al., 2009).

The bacterial and fungal community functions were predicted using FAPROTAX (Louca, Parfrey & Doebeli, 2016) and FunGuild (Nguyen et al., 2016) software, respectively. The nearest taxon index (NTI) was calculated for the microbial phylogenetic diversity using the null model-independent swap with 999 randomization runs and 1000 iterations, utilizing “Picante” package (v1.8.1) (Stegen et al., 2012) in R (v3.5.1).

Statistical analysis

To determine the significance of differences in microbial community composition between groups, non-parametric multivariate analysis of variance (PERMANOVA, transformed data by Bray–Curtis, permutation = 999) was performed using the adonis function of “vegan” package (v2.5-7) in R (v3.5.1). The Spearman’s correlations among microbial genera, environmental factors, putative pathogens, and the most abundant bacteria/fungi were analyzed with base functions in R (v3.5.1). A Mantel test (999 permutations) was applied to calculate the correlation between environmental factors and the microbial community using “vegan” (v2.5-7). Regression analyses were performed to investigate the relationships between the root rot ratio and biotic/abiotic factors (culturable microorganisms, soil properties, and alpha/beta-diversity indices). Statistical differences between treatments (soil properties, alpha diversity, NTI values) were calculated using a one-way analysis of variance followed by a Tukey’s HSD test (P < 0.05 was considered to indicate significant differences).

Results

Physicochemical properties of soil

The addition of BS (CH0d-CF0d) increased acetate and propionate contents in the soil (Fig. 1). Total VFAs were significantly higher in the BS-addition groups (t-test with P < 0.05: CH0d 282 ± 10 vs. Soil 7 ± 1; CF0d 582 ± 82 vs. Soil 7 ± 1). The levels of acetate and propionate gradually decreased to below the limits of detection (CH30d and CF30d) after 30 days, whereas butyrate levels increased and valerate was maintained at nearly constant levels.

Figure 1 Physicochemical properties of soil.

Colors from steel blue to orange represent the lowest to highest value (means ±SD: calculated with three replicates in each group) in each row, and values within the same row followed by different lowercase letters indicate significant differences among different treatments at P < 0.05 according to Tukey’s HSD test. “ND” means below the limits of detection.The names of treatment (group) abbreviation are defined in Table 1.

Following WF and BSF treatment for 15 or 30 days, pH and NH4+-N increased and Eh decreased as compared to in the initial soil (Fig. 1, CK15d or CK30 vs. Soil; CH15d or CH30 vs. Soil; CF15d or CF30 vs. Soil). These changes were significant in the BSF treatments (P < 0.05), which induced more conductive/anaerobic conditions (Eh <0 mV) compared to those in the WF treatment. Additionally, compared with those in the corresponding CK, pH and EC were higher, whereas Eh was lower in BSF-treated soils. This trend was more obvious at higher concentrations of BS (CF treatment, P < 0.05) after a longer treatment duration (Eh value: −68 mV in CH15d, −105 mV in CF15d; −88 mV in CH30d, −112 mV in CF30d).

Pathogenicity assay of soil

Pathogenicity (i.e., potential to cause root rot) of water or BS-treated soils on Sanqi roots was assayed, and the root rot ratio was calculated to evaluate the effects of different treatments. Symptoms of root rot typically included soft, necrotic, and brown roots (Fig. 2A). Healthy roots showed no symptoms and remained intact (Fig. 2B). Results indicated that both untreated continuous-cropping (“Soil”) and water-flooded (“CK”) soil showed the highest pathogenicity potential, as approximately 90% of the root showed root rot symptoms (Fig. 2C). In contrast, BSF treatment (CH and CF) significantly decreased (P < 0.05) the root rot ratios to as low as 10–20% (Fig. 2C).

Figure 2 Sanqi root rot ratio and number of culturable microorganisms in different soils.

(A) Rotted-symptom roots. (B) Healthy roots. (C) Root rot ratio in the initial soil (Soil), WF treated (CK), and BSF treated soils (CH and CF represent 50% diluted- and original BSF soils, respectively). (D) Number of culturable bacteria, fungi, Fusarium, and ratio of fungi/bacteria, ratio of Fusarium/fungi (up to down). Different lowercase letters indicate significant differences among different treatments at P < 0.05 according to Tukey’s HSD test. The names of treatment (group) abbreviation are defined in Table 1.

Population of culturable microorganisms in soil

Compared to untreated soil (Fig. 2D), 15 days of BSF treatment suppressed the population of culturable microorganisms, with a significant decrease in the fungal and Fusarium populations (P < 0.05). After 30 days, the fungal and Fusarium population further decreased slightly whereas the bacterial population returned to its original level. On day 30, the number of fungi was the lowest in CF, and the smallest Fusarium population was observed in CH. Moreover, the ratios of Fusarium/fungi and bacteria/fungi were both significantly reduced by BSF treatment, with CH treatment showing the lowest Fusarium/fungi ratio. In contrast, water-flooded treatment (CK) had no obvious effects on the culturable microorganisms tested (P < 0.05).

Alpha diversity of microbial community

After quality control, the sequences were clustered into 4,984 and 1,123 OTUs for bacteria and fungi, respectively. Alpha diversity analysis showed that the bacteria and fungi exhibited similar trends in the observed-species/Shannon indices after BSF (CH15d, CF15d, CH30d, and CF30d) and WF (CK15d and CK30d) treatments (Table 1). Specifically, the Shannon indices of both CH0d and CF0d decreased significantly compared with those in the original soil (P < 0.05). After incubation for 15 or 30 days, there was a notable increase in both indices in BSF-treated soils compared to in CH0d (or CF0d), although the values were lower than their corresponding CK values after the same duration. Additionally, both indices of diluted-BSF (CH, CH15d and CH30d) were slightly higher than those in the original BSF (CF, CF15d and CF30d) after the same treatment duration.

Structure of microbial community

PCoA and hierarchical cluster analysis were applied to compare the dissimilarities and hierarchical clustering of the microbial community between treatments, respectively. For bacteria, PERMANOVA showed a significant difference among all groups (R2 = 0.89, P = 0.001), and PCoA ordination demonstrated that the first two axes accounted for 81% of the variation (Fig. 3A). All samples were mainly separated into three clusters: “Soil-CK15d-CK30d” and “CH0d-CF0d” samples were separately clustered; the remaining BSF-treated samples (CH15d, CH30d, CF15d, and CF30d) formed the third cluster, in which “CH15d-CH30d” (CH, diluted-BS treated samples) and “CF15d-CF30d” (CF, original BS-treated samples) were slightly separated along the first axis, consistent with the results of hierarchical cluster analysis (Fig. 3C). A similar clustering pattern was found for fungi (PERMANOVA, R2 = 0.81, P = 0.001), but BSF-treated samples were more dispersed than the corresponding bacterial samples (Figs. 3B, 3D). Based on the different soil treatments and sample-clustering modes, the samples were combined to five larger groups for comparative analysis: (1) Soil (representing initial soil); (2) CHCF0d (CH0d and CF0d, representing BSF soils without incubation); (3) CK15.30d (CK15d and CK30d, representing WF soils incubated for 15 and 30 days); (4) CH15.30d (CH15d and CH30d, representing 50% diluted-BSF soils incubated for 15 and 30 days); and (5) CF15.30d (CF15d and CF30d, representing original BSF soils incubated for 15 and 30 days). To assess the statistical significance of differences in the bacterial and fungal communities between the five groups, differences between pairwise groups were examined using PERMANOVA, which revealed significant differences between groups, except for in “Soil vs. CK15.30d” (Table S1, bacteria: R2 = 0.45, P = 0.062; fungi: R2 = 0.27, P = 0.053).

Table 1 Alpha diversity of bacterial and fungal communities.

Group	Bacteria	Fungi	
	Observed-species	Shannon	Observed-species	Shannon	
Soil	2,544 ± 63 a*	9.49 ± 0.13 a	615 ± 48 a	6.49 ± 0.12 a	
CH0d	1,256 ± 171 c	5.62 ± 0.36 c	250 ± 56 d	2.47 ± 0.32 c	
CF0d	1,186 ± 41 c	5.76 ± 0.18 c	215 ± 25 d	2.28 ± 0.15 c	
CK15d	2,357 ± 257 ab	9.26 ± 0.14 a	437 ± 15 bc	6.02 ± 0.16 a	
CH15d	1,650 ± 128 bc	7.62 ± 0.10 b	287 ± 98 cd	4.39 ± 1.10 b	
CF15d	1,416 ± 106 c	7.12 ± 0.21 b	206 ± 8 d	3.58 ± 0.22 bc	
CK30d	2,274 ± 393 ab	9.28 ± 0.44 a	489 ± 41 ab	6.00 ± 0.06 a	
CH30d	1,441 ± 170 c	7.52 ± 0.28 b	258 ± 59 d	4.18 ± 0.86 b	
CF30d	1,176 ± 391 c	6.59 ± 0.88 bc	257 ± 67 d	3.85 ± 0.01 bc	
Notes.

* Values (means ± SD: calculated with three replicates in each group) within the same column followed by different lowercase letters indicate significant differences among different treatments at P < 0.05 according to Tukey’s HSD test.

Group abbreviations Soil initial soil, untreated

CH0d 50% diluted-BSF soils without incubation

CF0d original BSF soils without incubation

CK15d WF soils incubated for 15 days

CH15d 50% diluted-BSF soils incubated for 15 days

CF15d original BSF soils incubated for 15 days

CK30d WF soils incubated for 30 days

CH30d 50% diluted-BSF soils incubated for 30 days

CF30d original BSF soils incubated for 30 days

Figure 3 Principle coordinated analysis (PCoA) and hierarchical cluster analysis.

(A) Bacterial PCoA. (B) Fungal PCoA. (C) Hierarchical cluster analysis of bacteria. (D) Hierarchical cluster analysis of fungi. Both analyses are based on Bray–Curtis distance matrix. The dotted lines in (A) and (B) indicate the distribution of samples in each group (Soil, CHCF0d, CK15.30d, CH15.30d, or CF15.30d). Group name stands for: Soil: initial soil, untreated; CHCF0d: BSF soils without incubation; CK15.30d: WF soils incubated for 15 and 30 days; CH15.30d: 50% diluted-BSF soils incubated for 15 and 30 days; CF15.30d: original BSF soils incubated for 15 and 30 days.

Composition of fungal community

Fungal OTUs were classified into 10 phyla and 190 genera. Of these, one OTU (OUT_1, only classified at the kingdom level, Dataset S1) that was not present in the initial soil showed a notably high abundance in CH0d (63%) and CF0d (66%) compared to the other groups. After incubation for 15 or 30 days, the abundance of OTU_1 declined to 10% in CH15.30d and 14% in CF15.30d. Overall, BSF significantly modulated the composition of the fungal community from the phylum to the genus levels. Specifically, Ascomycota and Basidiomycota could be classified across all samples (Fig. 4A). Ascomycota were highly enriched in Soil and CK15.30d (67–75%) but were depleted in CH15.30d (11–15%, metastats analysis with P < 0.05). At the genus level, six genera that presented average relative abundances above 1% were Fusarium, Chaetomium, Staphylotrichum, Setophoma, Humicola, and Saitozyma. Of these, Fusarium was dominant across all samples (Fig. 4B) and was significantly depleted in CH15.30d soils compared with that in CK15.30d and CF15.30d soils (P < 0.05). The other five genera were significantly decreased after BSF treatment (P < 0.05). LEfSe analysis indicated that some Ascomycota affiliations (Sordariales, Staphylotrichum, Setophoma, and Fusarium) declined markedly in CH15.30d compared with those in CK15.30d soils (P < 0.05; Fig. 5F).

Figure 4 Taxonomic composition of the fungi and bacteria in different treatments.

(A) Fungi at the phylum level. (B) Fungi at the genus level. (C) Bacteria at the phylum level. (D) Bacteria at the genus level. Each bar represents the average relative abundance of a taxon in a treatment. “Others” in (A) and (C) represents the sums of the remaining fungal or bacterial phylum except for the top 10 abundant phylum, and “Others” in (B) and (D) represents the sums of the remaining fungal or bacterial genus except for the top 20 abundant genus. “unidentified” means unclassified, which was followed by the closest taxon at a particular classification level. The names of treatment (group) abbreviation are defined in Table 1 and Fig. 3.

Figure 5 Comparison of the microbial compositon and structure in different treatments.

(A, B) Venn diagrams display the number of unique and shared OTUs between the initial soil and CHCH0d in the fungal (A) and bacterial (B) communities. (C) Comparison of the relative abundance of the four potential fungal pathogens, the point size and number represent the average relative abundance (%). (D) Nearest taxon index (NTI) analysis, different letters indicate significant differences among groups at P < 0.05 according to Tukey’s HSD test. (E, F) LDA effect size analysis (LEfSe) among different microbial groups, the group-enriched taxa are displayed using ternary diagram. The names of treatment (group) abbreviation are defined in Table 1 and Fig. 3.

Furthermore, 219 fungal OTUs were common to both Soil and CHCF0d or unique to CHCF0d (Fig. 5A). According to previous studies (Li et al., 2020; Liu, Yang & Zhu, 2019; Miao et al., 2006), combined with the taxa detected in our study, four taxa (Fusarium, Cylindrocarpon, Alternaria, and Phoma) were potential fungal pathogens of Sanqi root rot. Next, we analyzed changes in the relative abundance of these fungi in response to the BSF and/or WF treatments (Fig. 5C). Compared with those in the initial soil, the levels of these four genera decreased by 4–200-fold after BSF treatment (CH15.30d and CF15.30d), except for a slight increase in Fusarium in the CF15.30d group (21.169% in Soil, 28.429% in CF15.30d). In contrast, the abundance of Fusarium, Cylindrocarpon, and Phoma was slightly increased in WF soils (CK15.30d) compared to in the initial soil.

Composition of bacterial community

Prokaryotic OTUs were classified into two kingdoms (Archaea and Bacteria), 64 phyla, and 608 genera. The most abundant Archaea OTU (phylum Euryarchaeota; genus Methanocorpusculum) was present in CF30 (0.2%). The five dominant phyla, which accounted for 82% of the bacterial community across all samples, were Proteobacteria, Bacteroidetes, Firmicutes, Synergistetes, and Acidobacteria (Fig. 4C). After BSF treatment, the relative abundance of Proteobacteria, Actinobacteria, and Acidobacteria decreased (from 46% to 18–40%, 11% to 0.4–2%, and from 10% to 0.5–2%, respectively), whereas that of Firmicutes, Synergistetes, and Bacteroidetes increased significantly (P <0.05; from 0.9% to 10–16%, from 0.04% to 3–14%, and from 11% to 22–49%, respectively). At the genus level, the top 10 genera present at more than 1% abundance were unidentified Rikenellaceae, Trichococcus, Macellibacteroides, Lactobacillus, Arcobacter, unidentified Synergistaceae, Proteiniphilum, Sphingomonas, Pseudomonas, and Bacteroides (Fig. 4D). LEfSe analysis showed that the phyla Firmicutes (affiliated order Clostridiales; family Ruminococcaceae; genus Anaerovorax), Synergistetes (genus unidentified Synergistaceae), and Bacteroidetes (genus unidentified Rikenellaceae, Macellibacteroides, and Proteiniphilum) were consistently and significantly enriched following BSF treatment (P < 0.05). Actinobacteria (genus Bryobacter), Proteobacteria (genus Sphingobium and unidentified GammaProteobacteria), and Gemmatimonadetes (family unidentified Gemmatimonadaceae) were the marker taxa (identified by LEfSe) in the initial soil (Fig. 5E). Among the top genera, unidentified Rikenellaceae, Macellibacteroides, Lactobacillus, unidentified Synergistaceae, Proteiniphilum, Pseudomonas, and Sedimentibacter were specific to the CH0d and CF0d groups compared to in the initial soil. Furthermore, the abundance of some bacterial genera differed significantly between groups; the relative abundance of Trichococcus, Lactobacillus, and Arcobacter was significantly higher in CH0d-CF0d than that in other groups (P < 0.05). Additionally, there were no significant differences between the initial soil and CK15.30d at either the phylum or genus level (Figs. 4C, 4D).

Comparing the bacterial OTUs between the initial Soil and CHCF0d soils revealed that 466 OTUs were unique to CHCF0d (Fig. 5B). These OTUs are mainly facultative or anaerobic bacteria (unidentified Rikenellaceae, unidentified Synergistaceae, and Sedimentibacter).

NTI analysis of the microbial community

NTI analysis of the bacterial and fungal samples revealed that all NTI values were higher than zero (Fig. 5D), and the average NTI value (3.22 ± 0.78) of the bacterial community was significantly higher (P < 0.05) than that of fungi (1.91 ± 0.58), providing evidence for phylogenetic assembly, particularly in bacteria. After BS amendment (CH0d and CF0d), the bacterial NTI decreased significantly compared with that in the initial soil (P < 0.05). The bacterial NTI of CH0d increased after 15 days (CH15d) and continued to increase after 30 days (CH30d), whereas no significant changes were observed under the original BSF or CK treatment. Furthermore, the NTI of the CH treatments was higher than that of the corresponding CF treatment, particularly after 30 days. A similar trend was observed for the fungal NTI, except that the NTI of CH30d was lower than that of CH15d.

Function analysis of microbial community

The functions of the bacterial OTUs were annotated using FAPROTAX. All BSF-treated samples harbored enriched functions involved in anaerobic metabolism, such as “methanogenesis”, “sulfate_respiration”, and “methanotrophy” (Fig. S1A), which was consistent with the increased abundance of anaerobic bacteria (e.g., Methanosaeta, and Desulfovibrio).

Fungal functional prediction using FunGuild showed that functions related to “plant pathogen” were more enriched in the Soil and CK15.30d samples than in the CH15.30 and CF15.30d samples (Fig. S1B).

Correlation between the potential pathogens and top abundant taxa

Spearman’s correlation analysis between the four potential pathogens and most abundant (top 20) bacteria, as well as fungi at the genus level, revealed that the four pathogens were significantly negatively correlated with most bacteria; however, they were significantly positively correlated with most fungi (Figs. 6A, 6B). In addition, correlation analysis between the top 25 abundant bacteria and fungi revealed a higher proportion of negative (71%) than positive (29%) correlations (Fig. 6C).

Figure 6 Spearman’s correlations between the top bacteria/fungi and potential pathogens, and with the environmental factors.

(A, B) The Spearman’s correlations between the top bacteria (A) or fungi (B) and the potential pathogens. (C) The Spearman’s correlation between the top bacteria and fungi, and with the environmental factors. The legend value (with corresponding colors from steel blue to red) is the correlation coefficient, the asterisks (* and **) indicate significant correlations at P < 0.05 (and 0.01). The colors of links between fungi/bacteria and environmental factors represent significantly (P < 0.05) negative (blue) or positive (brown) correlations.

Correlation between microbial community and soil physicochemical properties

As soil condition shape indigenous microbiota (Liu et al., 2019), we used Eh, and EC, pH, NH4+-N, and VFAs as environmental factors to clarify the relationships between the microbiota and soil properties. The Mantel test showed that environmental factors were significantly correlated with changes in the bacteria (r = 0.55, P = 0.001) and fungi (r = 0.42, P = 0.002). In addition, Spearman’s rank correlation analysis was used to evaluate the correlations between soil properties and bacteria or fungi at the genus level. Among the top 25 genera, most bacteria were significantly negatively correlated with Eh (P < 0.05) but positively correlated with EC, NH4+-N, and VFAs (acetate, propionate, and valerate, Fig. 6C).

Most fungi were significantly negatively correlated with EC, pH, NH4+-N, and VFAs (P < 0.05), and positively correlated with Eh. Particularly, the potential pathogens (Fusarium, Cylindrocarpon, and Phoma) were significantly negatively correlated with EC, pH, NH4+-N, and all tested VFAs (P <0.05).

Additionally, variation partitioning analysis (Fig. S2) revealed that the soil physicochemical properties, treatment modes (BS application or not), and treatment days explained 79% and 69% of the observed variation in the bacterial and fungal compositions, respectively. Soil properties, which were clearly affected by the treatment mode, explained 20% and 31% of the observed variation in the bacterial and fungal compositions, respectively, whereas treatment days only explained a small portion of the variation (2% and 3%, respectively).

Correlation between root rot ratio and biotic/abiotic factors

Linear regression analyses (Fig. S3) showed that root rot ratio was significantly (P < 0.05) positively correlated with the number of culturable fungi and Fusarium, ratio of Fusarium to fungi, ratio of fungi to bacteria, alpha diversity (Shannon index) of the bacterial and fungal communities, and Eh value in the soil. Root rot ratio was negatively correlated with the number of culturable bacteria (P = 0.16), concentrations of total VFAs (P = 0.056), EC (P < 0.05), and NH4+-N (P < 0.05). Moreover, root rot ratio was significantly correlated with the bacterial and fungal beta-diversity index (PCoA1) (P < 0.05).

Discussion

Maintaining soil fertility and controlling soilborne diseases are vital for sustainable crop production. BS, an organic fertilizer with high quality and biological activity, is an excellent candidate for green agriculture (Cao et al., 2016; Insam, Gómez-Brandón & Ascher, 2015; Walsh et al., 2012). We investigated the impacts of BSF application on the microbial community (including potential pathogens) and occurrence of root rot symptoms in Sanqi continuous soil.

BSF improved soil conditions

BSF has similar characteristics as ASD; both introduce organic materials into soil and create reductive/anaerobic conditions (i.e., in this study, mesocosms were flooded and sealed to reduce oxygen availability). Mechanisms underlying the suppression and efficacy of ASD include strong anaerobic/reductive conditions, changes in the microbial population, and production of organic acids and ammonia (Huang et al., 2015b; Momma, Kobara & Momma, 2011). ASD was recently adopted to control Sanqi soil disease and alleviate replant failure (Li et al., 2019). Considering the procedures used to apply these agents, the mechanisms through which ASD suppresses pathogens may also be partially applicable to BSF.

Firmicutes-affiliated anaerobes (Clostridiales and Ruminococcaceae, frequently enriched in response to ASD treatment) that produce toxic VFAs (Huang et al., 2015a; Momma et al., 2013; Mowlick et al., 2013), were significantly enriched by BSF treatment. High concentrations of VFAs can suppress pathogens in ASD-treated soil (Momma et al., 2006). Both BS treatments incorporated a large amount of BS-derived VFAs (acetate and propionate) into the initial soil, and the persistence of butyrate generated in the soils showed the potential to suppress pathogens throughout the experiment. This result was supported by the negative correlation between pathogens and VFAs. Notably, many facultative and anaerobic taxa (such as unidentified Rikenellaceae, Trichococcus, Macellibacteroides, and Lactobacillus), which were unique to CHCH0d and derived from BS, were incorporated into the initial soil by BSF. Furthermore, some of these taxa were highly abundant throughout BSF treatment (30 days). These taxa, in turn, consume residual oxygen and create more anaerobic conditions, further suppressing pathogens (Wen et al., 2015).

Most fungi (including three potential pathogens) were positively correlated with the Eh value, indicating they were suppressed by the anaerobic/reductive conditions. This is consistent with the observation that most fungi cannot grow under anaerobic conditions (Takaya, 2002). Furthermore, the increased pH and NH4+-N from BSF may mitigate soil acidification in the Sanqi cropping system. Therefore, soil pre-conditioned with BSF presented better global physicochemical properties and higher quality than the initial soil, which may contribute to synergistic inactivation of pathogens. However, application of BSF is not always practical, particularly in hilly mountain areas. Hence, irrigating soil with BS to 100% or less of the water holding capacity, similar to previous studies (Cao et al., 2016; Wen et al., 2015), may be an alternative approach for suppressing pathogens.

In contrast, compared with those in the BSF-treated soils or initial soil, no significant changes in either the bacterial or fungal communities were observed in the WF treatment. Furthermore, there was no obvious suppression of potential pathogens, including the culturable Fusarium, or a reduction in the root rot ratio. This result is inconsistent with those of previous studies showing that WF can effectively control pathogens (Kelman & Cook, 1977; Niem, Gundersen & Inglis, 2013). Thus, WF may not lead to significant changes in the microbiota or suppression of pathogens, emphasizing the key role of anaerobic microorganisms and organic matter derived from BS.

BSF reshaped the soil microbial community to a relatively pathogen-suppressive state

BSF reshaped the bacterial and fungal community to form a different community structure (BSF-treated groups clustered together, as shown in Fig. 3), which was consistent with Cao et al. (2016). Importantly, BSF treatment reduced the abundance of potential pathogens (Fusarium, Cylindrocarpon, Alternaria, and Phoma), which might be the key factor in reducing the root rot ratio of Sanqi. Fusarium, one of the main pathogens, was significantly reduced to a similar number by both CH and CF treatments according to the plate count assay (Fig. 2D). However, ITS sequencing showed that relative abundance of Fusarium was reduced following CH treatment but increased following CF treatment (Fig. 4B). Indeed, differences exist between culturable and culture-independent sequencing methods. The former is mainly affected by number of culturable microorganisms detected in a specific medium, the latter is primarily affected by the DNA extraction method, primers used to amplify the 16S or ITS genes, and sequence data analysis method (Bonk et al., 2018). We counted the number of Fusarium using Komada’s selective medium, which has been used in previous studies (Tao et al., 2020; Wen et al., 2015); Tao et al. (2020) showed that quantitative colony counting was positively correlated with the results of qPCR. One possible reason for this is that CF treatment induced more anaerobic/reductive conditions (lower Eh value) compared to CH, suppressing more total fungi, whereas Fusarium which has a higher tolerance was less affected (Ebihara & Uematsu, 2014), leading to an increased ratio of Fusarium to fungi in CF compared with that in CH (Fig. 2D). The suppression of other fungi, except for Fusarium, was greater following CF, resulting in a higher relative abundance of Fusarium in CF than in CH or Soil (Fig. 2D). The root rot ratio was positively correlated with the culturable Fusarium and ratio of Fusarium/fungi, supporting that a reduced Fusarium/fungi ratio is conducive to the growth of Sanqi (Zhao et al., 2017). Notably, taxa classified based on OTUs are generally accurate at the genus level, and some non-pathogenic Fusarium spp. contribute to the abundance but not to the occurrence of disease. As only culturable fungi can be enumerated using the plate count method, further absolute quantification methods, such as qPCR can be applied to verify the results. Additionally, we observed a slightly negative correlation between the root rot ratio and culturable bacteria number, which is consistent with a previous study showing that the total number of culturable bacteria is important for disease suppression (Bonanomi et al., 2010).

Overall, although potential pathogens in the soils were not eliminated, they were suppressed by BSF. This result was supported by functional prediction of fungi, which showed fewer “plant pathogen” and pathogenic potential in BSF-treated soils.

BSF shifted bacterial community to harbor more beneficial taxa

Compared with the initial soil, BSF treatment significantly altered the bacterial community: Synergistetes, Firmicutes, and Bacteroidetes were enriched, whereas Actinobacteria and Proteobacteria were depleted. Members of Synergistetes can participate in synergistic acetate oxidation and organic acids digestion to produce substrates for hydrogen methanogens, which play important roles in anaerobic metabolism (Acs et al., 2015). Bacteroidetes are beneficial taxa that are abundant in the rhizosphere of wild plants; in contrast, members of Actinobacteria and Proteobacteria are frequently enriched in soils with pathogen infection and long-term N fertilizer application, and are considered as marker taxa of poor soil quality (Dai et al., 2018; Perez-Jaramillo et al., 2018; Wu et al., 2016). A similar study showed that taxa of Proteobacteria and Actinobacteria are highly enriched in Sanqi monoculture soil (Zhao et al., 2017). Furthermore, Wu et al. (2016) considered Proteobacteria as a marker taxon in Sanqi root rot soil. The decreased abundance of Proteobacteria and Actinobacteria may reflect the positive effect of BSF treatment and elevated soil micro-conditions against plant disease.

Under BSF treatment, the most enriched genera (e.g., unidentified Rikenellaceae, unidentified Synergistaceae, and Sedimentibacter) were mostly facultative/anaerobic, which is consistent with a previous study (Chen et al., 2012). Highly abundant anaerobic taxa have also been observed in maize-Sanqi rotation systems (Zhao et al., 2017), representing improved soil quality against disease. Furthermore, archaeal microorganisms (Methanocorpusculum and Methanosaeta), although present at relatively low abundance, were enhanced. These facultative/anaerobic taxa may compete with pathogenic fungi for niches.

Bacterial community was vital for defending against pathogens

Plant roots interact with many microorganisms, including bacteria, fungi, and oomycetes in soil, where fungi and oomycetes can typically cause serious disease. Bacterial community often negatively correlate with eukaryotic microbes and defend against virulent root-related eukaryotes (Duran et al., 2018; Luo et al., 2020). Similarly, we demonstrated that culturable bacteria were slightly increased but culturable fungi were significantly suppressed under BSF treatment at day 30, which is similar to a previous study (Walsh et al., 2012). Moreover, Spearman’s correlation analysis revealed greater negative correlations between the top 25 abundant bacteria and fungi, as well as significant negative correlations between numerous bacteria and potential fungal pathogens. This indicates a potentially competitive or antagonistic relationship between these two groups. This, combined with the negative correlation between the number of culturable bacteria and root rot ratio, suggests that the bacterial community plays a vital role in suppressing fungal pathogens and controlling disease. However, this study was conducted in microcosms, without considering the interactions between plant roots and microbiota in the field. Pathogen levels may re-increase following planting, possibly because of the stimulation of specific root exudates (Li et al., 2014; Liu et al., 2018). Considering that pathogen resurgence is a risk after planting, this issue may be mitigated by introducing antagonistic bacteria alongside BS application (Yin et al., 2021).

Culturable microorganisms can be used as indicators of soil pathogenicity

The root rot ratio was positively correlated with the alpha diversity (Shannon index) of the bacterial and fungal communities. Higher microbial diversity is vital for community stability and pathogen suppression (van Elsas et al., 2012). However, this association is controversial; microbial diversity, e.g., relative abundance, may not be a credible indicator of plant health (Huang et al., 2019; Xiong et al., 2017). In some cases, a community with higher microbial diversity can harbor a lower biomass (i.e., the absolute number of microbes in a certain environment) (Chen et al., 2017), which is consistent with the present study, as shown in the initial soil vs. CH0d (or CF0d) (harboring more microorganisms from BS). In the present study, the highly abundant fungus OUT_1 in CH0d and CF0d caused unevenness in the community and, as expected, led to a decrease in the alpha diversity indices. Alternatively, some absolute quantification measures, such as the counts of culturable microorganisms in this study or combined with qPCR (Tao et al., 2020), may partially reflect the real diversity of the community and serve as a credible indicator of disease suppression.

Conclusions

We created a conceptual graph describing the effects of BSF application on soil properties and the microbial community structure (Fig. 7). The anaerobic/reductive environments created by BSF treatment elevated the soil physicochemical properties (e.g., pH, NH4 +-N, and Eh) and strongly influenced the bacterial community. Following BSF treatment, bacteria were enriched and fungi were suppressed. In addition, the phylogenetic relatedness of the bacterial community was higher than that of fungi community (NTI analysis). Together, these factors were responsible for assembling a stable bacterial community that negatively interacted with the fungal community, possibly through antagonism or competition for nutrients and niches, thereby suppressing the fungal pathogens and alleviating root rot. This study provides a valuable reference for the application of BS, which may contribute to alleviating root rot disease in Sanqi production.

Figure 7 Conceptual graph describing the effects of BS application on soil properties and the microbial community structure (including potential fungal pathogens).

Supplemental Information

Supplemental Information 1 Statistical significance of the differences between pairwise groups in the microbial community using PERMANOVA

Pairwise Groups: The two groups applied for assessing statistical significance of differences in bacterial or fungal composition among different pairwise groups using PERMANOVA (Bray–Curtis, permutation = 999); R2: Interpretation degree of difference; P value and Significance: Reliability of the statistical test (P value), “-” indicates P ≥ 0.05, “*” indicates 0.01 P < 0.05, “**” indicates 0.001 < P ≤ 0.01, and “***” indicates P ≤ 0.001.

Click here for additional data file.

Supplemental Information 2 Heatmap illustrating the clustering and relative abundance of predicted functions in bacterial and fungul communities across all samples

(A) Bacteria. (B) Fungi.

Click here for additional data file.

Supplemental Information 3 Variance partitioning analysis (VPA) of the effects of soil properties, treated modes, treated days, and the interactions of those factors on the bacterial and fungal community

(A) Bacteria. (B) Fungi.

Click here for additional data file.

Supplemental Information 4 The linear regression relationships between the root rot ratio and the soil properties, number (or ratio) of culturable microorganisms, and alpha/beta-diversity indices

Click here for additional data file.

Supplemental Information 5 Sequence of fungal OTU 1

Click here for additional data file.

Supplemental Information 6 Raw data of Fig. 2

Click here for additional data file.

Supplemental Information 7 Raw data of Fig. 2D

Click here for additional data file.

We gratefully acknowledge the anonymous reviewers and editors for their valuable comments and suggestions that have greatly improved this manuscript. We thank Dr. ShuBo for discussions on this manuscript. We also thank Editage for language editing.

Additional Information and Declarations

Competing Interests

Author Contributions

Data Availability

The authors declare there are no competing interests.

Chengxian Wang conceived and designed the experiments, performed the experiments, analyzed the data, prepared figures and/or tables, authored or reviewed drafts of the article, and approved the final draft.

Jianfeng Liu analyzed the data, prepared figures and/or tables, authored or reviewed drafts of the article, and approved the final draft.

Changmei Wang analyzed the data, authored or reviewed drafts of the article, and approved the final draft.

Xingling Zhao analyzed the data, authored or reviewed drafts of the article, and approved the final draft.

Kai Wu analyzed the data, authored or reviewed drafts of the article, and approved the final draft.

Bin Yang analyzed the data, authored or reviewed drafts of the article, and approved the final draft.

Fang Yin analyzed the data, authored or reviewed drafts of the article, and approved the final draft.

Wudi Zhang conceived and designed the experiments, prepared figures and/or tables, authored or reviewed drafts of the article, and approved the final draft.

The following information was supplied regarding data availability:

The sequencing data generated in this work is available at NCBI Sequence Read Archive: PRJNA661430 and PRJNA661668, for the bacterial and fungal sequences, respectively.

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
