# Peer review of "Biogas slurry application alters soil properties, reshapes the soil microbial community, and alleviates root rot of Panax notoginseng"

_PeerJ, doi:10.7717/peerj.13770_

## Round 0.1 · original submission · Major Revisions

We received two detailed and constructive reviews. I agree with them - the manuscript requires major revisions. I made a number of suggestions for style on the first 60 lines. These are not comprehensive. Please use those suggestions as examples.

Other general points.

1. Provide a university or institution email address for the corresponding author, unless the author is no longer associated with a school or institute.

2. In background for Abstract describe why ginseng root disease is important. In other words, it is not clear why this application of BSF was investigated. Abstract should include Conclusion.

3. Acknowledge funding sources.

4. Deposit fastq files into the SRA and provide an accession number in methods.

5. Replace bar charts in Figure 3 with box plots (line figure 5d) or violin plots.

Specific Comment Examples

line 21. Write with active voice. Revise to “..(WF) can control…”
Line 22-25. This sentence runs on and needs revision. Start by stating what is known about BSF, then state a hypothesis and predict a treatment effect.
Line 26-29. Again, this is a run on sentence. Start with “In this study, we conducted a microcosm experiment to determine if BSF can control… Microcosms contained soil collected from…”
Line 33. Delete “Results indicated that the”
Line 34. Delete “the” whenever possible. See line 52, 54
Line 35. Keep sentence structures parallel (reshaped…, reduced… and suppressed…)
Line 49. Revise to “Biogas production by anaerobic digestion of human and animal waste, straw, and other organic material has emerged as a potentially important source of renewable energy..”
Line 54. Delete “The sustainability…environment”
Line 57. Write succinctly. Revise to “Application of BS improves soil quality and enhances crop yield..”
Line 62. Here and throughout. Delete phrases like “has been shown to” and “Previous studies have shown that (line 76).

Reviewer 1 ·

Basic reporting

• It might be helpful for the authors to provide more specific details about why biogas slurry is a suitable fertilizer. What is the C:N? Do all biogass slurries have similar compositions? Which volatile organic compounds are found in BS, and why do those impact biological activity?
• What crops are used for the rotation strategy? (L85)
• Additional details on the previous results of BSF might also be helpful for the reader. For example, how and/or why does BSF promote nutrient cycling? Which nutrients? Does BSF have positive impacts for all soil types and/or environmental conditions?
• Related to the previous point, it is not completely clear why the authors decided that BSF and WF might specifically help with Sanqi root rot. Did previous studies find that BSF and/or WF controls other Fusarium spp.?
• It would be helpful for the authors to provide some hypotheses about their treatments.
• For figures and tables, it would be helpful to provide information reminding the reader what the sample labels stand for (i.e. CK1d = Check 1 day, etc.). In general, figure captions need a little more information describing the figure. For example, what statistical analysis was used for Fig 1? In Fig 3a and b, what do the dotted lines indicate?

Experimental design

• Why was a 1:1 soil:water ratio chosen for the WF treatment? Why dilute BS by 50% and not 25% or some other percentage?
• How much soil was collected at each sampling event?
• If the bottles were sealed, was there any additional oxygen provided during the incubation period? If gas samples weren’t being collected, why were the bottles sealed? Were they incubated in the dark or light?
• Were samples collected from the same bottles for each time point? If so, did the unsealing of the bottles change the air conditions?
• Was there a no-treatment control, i.e. a soil that was not flooded?
• How was quality filtering of the raw sequencing reads performed?
• What p-value was considered significant?
• Why was NH4+ the only nutrient assessed? Why not total N or carbon?

Validity of the findings

• Were the differences in VFAs with the addition of BS statistically significant? If so, that should be clearly indicated in the Results, and p-values should be presented for non-significant differences.
• As there were multiple time points, it is not clear in L242 to when pH and NH4+ increased and Eh decreased. Be specific as possible when reporting results.
• Rather than describe the clustering of the samples in the PCoA, it would be more informative for the authors to use the statistical results to indicate which groups had significantly different community compositions.
• It is not clear what the authors mean by the samples being “regrouped” into 5 larger groups (L292). Were these groups analyzed independently?
• L305: What do the authors mean by “globally”?
• The pathogenicity of many fungi is often at the species level. However, sequencing generally only provides genera-level identification. While using changes in the abundance of fungal genera with known pathogens to assess treatment efficacy is a good step, the authors should acknowledge the limitations of these results and analyses.
• Bacteria and fungi taxonomic information are not always properly written. There should not be any underscores in an organism name.
• What do the authors mean by “marker taxa”? How were marker taxa identified?
• L423: soil organic matter was not measured, so how can the authors be certain that BSF treatment “introduced large amounts of organic materials into the soil”?
• How do the authors know that the fungus labeled “OTU_1” is a facultative and/or anaerobic organism?
• There are many taxa within the Proteobacteria and Actinobacteria phyla that may be beneficial. By how much were Proteobacteria and Actinobacteria depleted compared to the control soils?
• L539: What do the authors mean that “a higher microbial diversity community can harbor the lower biomass”? Lower biomass of what?

Additional comments

• This manuscript provides the results of a lab incubation study examining the impact of using a biogas slurry to flood soils and control pathogens problematic for Sanqi growth. As the authors point out, this is similar to the anaerobic soil disinfestation technique that has been used to control similar pathogens in other cropping systems. The use of biogas slurry is an interesting idea and the results of this study provide excellent initial information about the potential microbial mechanisms by which biogas slurry flooding impacts pathogen abundance.
• However, some re-arranging and editing of the Introduction and Discussion would better help explain the rationale for the study. In particular, the information about ASD and the similarities to biogas slurry flooding should be provided in the Introduction.
• The authors do point out some of the limitations with a few of the methods (i.e. culturable methods), but do not address the limitations of performing lab incubations in sealed glass containers. Lab incubation studies are an acceptable method for initial studies, but the authors should point out the limitations and maybe provide more information regarding how the results found in this study may vary to field-based tests.

Reviewer 2 ·

Basic reporting

No comment.

Experimental design

No comment.

Validity of the findings

No comment.

Additional comments

Soil sterilization with biogas slurry and cultivation tests with plants have been conducted, and the contents are highly influential. However, before accepting the manuscript, the following needs to be addressed.


L195:
Is the software used QIIME? The software name and version should be provided.


L251: as approximately 90% of the root showed root rot symptoms (Fig. 2b).
Fig.2b does not show 90% of the root rot symptoms. This problem can be solved by changing the location of (Fig.2b) in the text.

Fig.2b and Fig.2c:
It would be easier for the reader to understand if the criteria for root rot were also shown. It is not clear which are normal and which are root rot.

L254:
"Notably, water treatment (CK) had no obvious effect on the pathogenicity of soil (Fig. 2a)." is not necessary; it duplicates the content of L250-252.

L275-278:
"CH15d/CH30d" and "CF15d/CF30d" are not intended to represent fractions, so the notation without "/" is preferred.


Table 1:
There is no "CK0d" in Table 1. Is it unmeasured?

L296:
In Fig. 3b, the plots of CF15.30d and CH15.30d overlap, is there a significant difference between them?

Fig.3b: 
Isn't CHCF1d a mistake for CHCF0d?


L300-L303:
In which figure is otu_1 located?
If there is no figure to refer to, insert "data not shown".

L299-352
L299-352 discusses the changes in the composition of bacteria and fungi due to biogas slurry treatment. However, a correct evaluation of biogas slurry treatment cannot be made without the addition of data on whether the "number" of plant pathogenic fungi such as Fusarium is reduced or not. Therefore, it is desirable to conduct real-time PCR (quantitative PCR) and resubmit the results with the data. (For example, the percentage of Fusarium in CF30d (Fig.4b) is apparently higher than that in control, but if the total number of fungi is reduced, then the number of Fusarium is actually reduced more than that in control.)

Fig2d suggests that the biogas slurry treatment reduced the fungal count. If it is not possible to perform real-time PCR (quantitative PCR), why not consider the results of Fig.2d (decrease in fungal and Fusarium counts) together with the results of Fig.4b?

---

## Round 0.2 · Minor Revisions

We received a review of the revised manuscript from reviewer 1 and I think this version addresses many of the comments of the second reviewer, who has declined to comment, and my earlier comments. I do think the manuscript needs a fresh read, after the suggested revisions are made, by a fluent English speaker.

I do have some specific comments and made a number of minor changes in the attached pdf.

- Use reasonable significant figures throughout. See text for numerous examples.

- The rebuttal letter has several points of discussion that should be made in Discussion. Readers will not have access to these rebuttals and will have similar concerns.

- I don't accept the reason for keeping Figure 2 in bar chart format. Also, the error bars are not shown. A boxplot with jitters of data points is appropriate.

Reviewer 1 ·

Basic reporting

L40: “more potential”

L45-48: This sentence needs to be revised. Possibly “Microcosms were established with soil from a P. notoginseng…” and then “followed by an investigation of changes…”

L55: “, suppressed the culturable fungi and Fusarium, and eventually decreased….”

L71: “…the number of biogas plants….”

L80: “…partly due to its biological activity….”

L82: “…reduced the incidence….”

L119: I don’t think “Meanwhile” is the appropriate transition word here. Maybe “In addition, ….)?

128: “….suppress pathogens and….”

L275: “…significantly increased in BS-addition groups….”

L291-2: “…while healthy roots showed no symptoms and remained intact….”

L390: While it is true that the Silva database identifies taxa as “unidentified_Rikenellaceae” (for example,) it is common practice to manually edit these results and just call them “unidentified Rikenellaceae.”

L428: “….most abundant bacteria….”

The authors have done a great job responding to the previous comments. The manuscript is much improved. Listed below are several grammatical/word-choice edits.

L477: “The present study showed…”

L481: “In the present study,…”

L490: “The present study…”

L499: “….similar to previous studies…”
L500: “…pathogens, but the efficacy in the field needs further….”

L525: “….ITS genes, and sequence data analysis method (Bonk…”

L528: “….higher tolerance were less effected…”

L529: “…increased ratio of Fusarium…”

L555: “…soils, and are considered marker taxa of….”

574: “…the soil environment,…but the bacterial….”

L577: “Similarly, the present study…”

L585: I’d replace “Additionally” with “However”

L592: “The present study…” In this paragraph, there are numerous places where sentences need to be edited to be “the present study” rather than just “present study.”

The Fig 3 caption has several typos.

Experimental design

No comment

Validity of the findings

No comment

---

## Round 0.3 · Minor Revisions

At this stage, since the revisions are minor, I don't think we will need another round of reviews. If appropriate revisions are made, I can make a decisions without further review.

Michael

Reviewer 1 ·

Basic reporting

The manuscript has greatly improved. There are still a few grammatical errors in the abstract and manuscript, primarily subject-verb agreement.

L47: It would be helpful to make sure that it is clear the authors are discussing changes to the soil bacterial and fungal communities and use that adjective more frequently to make this clear.

L55: What do the authors mean that BSF “promoted the bacterial community”? Which parts of the bacterial community?

L57: Do the authors mean BSF rather than just BS?

L273: Were the increases and decreases in pH and NH4+ statistically significant? If so, the authors should include this information. This information should be provided when reporting the results of all statistical analyses.

L456: What do the authors mean by BS being a fertilizer with “high quality”? Also, why would biological toxicity make an organic fertilizer a good candidate for green agriculture. To me, this phrase implies that it will be harmful to all biology, not necessarily just pathogens.

L602: What do the authors mean the “bacterial community was tightly clustered in the phylogenetic assembly”? What does a tight clustering indicate biologically?

Experimental design

No comment

Validity of the findings

No comment

Additional comments

The edits that need to be made are very minor. I do not think it needs to be sent out for review again if the editor can verify the corrections have been made.

Reviewer 2 ·

Basic reporting

No comment

Experimental design

No comment

Validity of the findings

No comment

Additional comments

This paper has been appropriately improved.
If the following points are appropriately addressed, it is considered acceptable.

Fig.4
Please add the criteria encompassed in the "Other" section to the figure's legend.

L462-464
In addition, correlation analysis between the most abundant bacteria and fungi revealed a higher proportion of negative (71%) than positive (29%) correlations (Fig. 6c).

Please add a definition of "the most abundant bacteria and fungi" somewhere.


L614-616
These facultative/anaerobic taxa may compete with pathogenic fungi or further create an anaerobic environment that is unfavorable for their survival.

Because methanogenic archaea utilize acetic acid and hydrogen, they do not compete nutritionally with pathogenic fungi. In addition, methanogenic archaea cannot consume oxygen and thus cannot create an anaerobic environment.
A reconsideration of the discussion is needed.

L627: most abundant bacteria and fungi
Please add a definition of "the most abundant bacteria and fungi" somewhere.


【Below, if possible.】
L326-327
Fusarium/fungi and bacteria/fungi were both significantly reduced by BSF treatment, with CH treatment showing the lowest Fusarium/fungi ratio

Why did the low concentration biogas slurry (CH) show better results than the high concentration biogas slurry (CF)? If possible, please add discussion.

---

## Round 0.4 · Minor Revisions

The reviewers' comments have been addressed but I found some minor edits, mostly for style, that should be addressed (see annotated *_v4. pdf). I think the paper is acceptable with these minor revisions I noted in the pdf.

---

## Round 0.5 · accepted · Accept

I appreciate your patience with the multiple revisions.

Regards,

Michael